# Using Real Options Thinking to Value Investment Flexibility in Carbon Capture and Utilization Projects: A Review

**Hanne Lamberts-Van Assche** [1,*] and **Tine Compernolle** [1,2]

1   Department of Engineering Management, University of Antwerp, Prinsstraat 13, 2000 Antwerp, Belgium; tine.compernolle@uantwerpen.be
2   Geological Survey of Belgium, Royal Belgian Institute of Natural Sciences, Jennerstraat 13, 1000 Brussels, Belgium
*   Correspondence: hanne.lamberts-vanassche@uantwerpen.be

**Abstract:** Carbon capture and utilization (CCU) is one of the key technologies that may help to reduce industrial emissions. However, the deployment of CCU is hampered by various barriers, including high levels of technical, policy and market uncertainty. The real options theory (ROT) provides a method to account for these uncertainties and introduce flexibility in the investment decision by allowing decisions to be changed in response to the evolution of uncertainties. ROT is already being applied frequently in the evaluation of renewable energy or carbon capture and storage (CCS) projects, e.g., addressing the uncertainty in the price of $CO_2$. However, ROT has only found a few applications in the CCU literature to date. Therefore, this paper investigates the specific types of uncertainty that arise with the utilization of $CO_2$, identifies the types of real options present in CCU projects and discusses the applied valuation techniques. Research gaps are identified in the CCU literature and recommendations are made to fill these gaps. The investment decision sequence for CCU projects is shown, together with the uncertainties and flexibility options in the CCU projects. This review can support the real options-based evaluations of the investment decisions in CCU projects to allow for flexibility and uncertainty.

**Keywords:** real options; carbon capture and utilization; carbon capture and storage; investment under uncertainty

## 1. Introduction

Mitigating climate change is one of the biggest challenges that humankind is facing in the 21st century. The search for low-carbon, or even carbon-negative, solutions to reduce $CO_2$ emissions is ongoing. Carbon capture and utilization (CCU) technologies can be part of these low-carbon solutions helping to address climate change. Whereas carbon capture and storage (CCS) technologies capture the $CO_2$ from a $CO_2$-emitting process and store it permanently underground, CCU technologies use the captured $CO_2$ as a resource to create valuable products or services [1]. Utilizing the $CO_2$ to create products generates additional revenues, thus lowering the net costs of reducing emissions [2]. Although the concepts of CCS and CCU are often intermingled, the rationale behind both technologies is completely different. CCS contributes directly to climate change mitigation by capturing and permanently storing $CO_2$ emissions underground. CCU, on the other hand, can help to reduce the dependency on fossil fuels by using already emitted $CO_2$ as a substitute and can play a role in the transition to using renewable energy systems [3]. The absence of a viable business case for CCS, due to its high costs and lack of incentives, has hindered its deployment. Contrary to CCS, CCU pathways could provide sufficient economic incentives through the cost savings from the reduction in fossil resources and the revenue from sold products [3].

In general, CCU technologies are classified into two broad categories: the direct use of $CO_2$ and the conversion of $CO_2$ [4]. Examples of the direct use of $CO_2$, where the $CO_2$

molecule is left unbroken, include enhanced oil recovery (EOR), refrigerant fluids or soft drink industries [4,5]. $CO_2$-EOR is in the gray zone between CCS and CCU: the $CO_2$ is injected into oil reservoirs to increase the production of oil (CCU) and is permanently stored in these reservoirs afterwards (CCS) [3]. This study includes $CO_2$-EOR as a CCU route for the sake of completeness. When it comes to $CO_2$ conversion, three broad categories are identified: mineralization, chemical-based conversion and bio-based conversion routes.

Hepburn et al. [2] estimate that CCU pathways could reach a total $CO_2$ utilization potential of 2.5 Gt of $CO_2$ per year by 2050. However, there are several challenges in taking $CO_2$ utilization to the market. The major challenge is the high stability of the $CO_2$ molecule, resulting in high energy requirements that are needed to break the bonds and convert the $CO_2$ [6]. Other challenges are the low technology maturity of CCU technologies, the lack of clear climate policies and regulatory frameworks for CCU, high investment costs, the need for green and cheap hydrogen and public acceptance of CCU [2,7]. These barriers hinder investments in CCU technologies, making it less likely that CCU projects will be scaled up soon. To investigate the economic feasibility of CCU projects, CCU researchers have resorted to techno-economic assessments (TEAs). A TEA integrates technical and economic feasibility evaluations into one systematic study [8]. The most common evaluation criteria in these TEAs for CCU is the net present value, which is based on the costs and revenue over the project's lifetime [8]. However, these traditional valuation methods do not consider the ability to adjust investment decisions or defer investment to a later phase [9]. Moreover, these traditional methods completely fail to capture the value of the additional flexibility that CCU installations may provide to existing plants, e.g., the ability to switch between energy sources, inputs or outputs. Hence, these traditional methods will likely underestimate the true value of CCU projects and will lead to sub-optimal investment decisions.

In this review article, an alternative evaluation method is presented, which recognizes the irreversibility and the flexibility of CCU investments: the real options theory (ROT). While traditional valuation methods only address uncertainties in a sensitivity analysis, ROT is based on the idea that projects or decisions can be changed in response to the evolution of uncertainties in the ever-changing world [10]. To evaluate investment decisions in low-carbon energy systems, real options theory is currently the most frequently used method for addressing the uncertainty in future revenue and costs and introducing flexibility [11]. Martinez-Cesena et al. [10] reviewed the real options studies for (renewable) electricity generation projects. The authors observed that ROT has the potential to increase the feasibility of these projects as it allows us to introduce and value flexibility in the investment decision. Schachter and Mancarella [12] provided a critical analysis on the application of ROT to value investment flexibility in smart grids and low-carbon energy systems. Ginbo et al. [13] reviewed the applications of ROT in investment decisions for climate change adaptation and mitigation projects and showed that ROT is particularly relevant for renewable energy projects because of their high risks and irreversibility. Kozlova [14] reviewed the existing ROT studies for renewable energy projects and observed a variety of real options models. This illustrates the need for a critical review of the real options methodology and the evaluation methods for renewable energy projects in general. Similar to renewable energy projects, CCU projects are also characterized by high uncertainty, risk and irreversibility of the investment. Hence, ROT is also highly relevant for CCS projects. Agaton [15] performed a bibliometric analysis, screening the CCS literature for real options applications. The literature search resulted in 67 studies, which were reviewed for the different types of uncertainties, options and valuation techniques that were applied for CCS projects.

The above-listed literature overview shows how ROT is already well developed and frequently applied in the evaluation of (renewable) energy projects and CCS projects. Real options methods are already adopted in 67 studies to value CCS projects, together with their flexibilities, uncertainties and risks [15]. Although CCU and CCS projects share some similarities, their main differences should be recognized as well: (1) with CCS, the $CO_2$ is

stored permanently, whereas the utilized $CO_2$ is only stored temporarily in CCU-based products; (2) CCS can store large quantities of $CO_2$, allowing for $CO_2$ capture from ambient air, while the demand for $CO_2$ in CCU pathways is limited by the demand for the products (chemicals, fuels); and (3) the economic incentive for CCS remains weak due to the high costs and the lack of revenue, whereas CCU projects create revenue by producing chemicals or fuels. Due to these differences, the evaluation of investment decisions in CCU projects through real options-based analyses can be significantly different from the real options-based studies for CCS projects. Nevertheless, the application of ROT to evaluate CCU investment decisions remains highly relevant because of the unique types of flexibilities, risks and uncertainties present in CCU pathways. Therefore, this review article aims to screen the existing ROT studies for CCU projects and explore the common sources of uncertainty, types of real options and valuation techniques. Section 2 presents the materials and methods for this review. Section 3 explores the general principles of real options analysis. In Section 4, the existing applications of ROT in the CCU literature are reviewed in detail. Section 5 discusses the remaining research gaps in the CCU literature and expresses recommendations to fill these gaps based on ROT studies in other research fields. The review article ends with a brief conclusion.

## 2. Materials and Methods

Before exploring the CCU research, in particular, the basic principles of real options theory were summarized. To understand the real options theory properly, the renowned handbooks of Dixit and Pindyck [16] and Trigeorgis [9] were consulted. The different methods used to value real options, sources of uncertainty and real options types are listed in Section 3.

Next, a literature review on the applications of ROT for novel CCU technologies was performed. Literature searches were completed in the Web of Knowledge and Scopus databases to retrieve all papers that performed a real options analysis in a CCU context. The first search query combined different variations of the term "carbon capture and utilization" and the term "real options". (the first search query was ("carbon capture and utili?ation" OR "$CO_2$ utili?ation" OR "$CO_2$ use" OR "carbon dioxide utili?ation" OR "CCU" OR "CCUS") AND ("real options")) These searches in the Web of Knowledge and Scopus database resulted in 10 and 11 papers, respectively. Seven duplicates were identified and removed and four more papers were deleted because they were out of scope (one was a review paper and three other papers were deleted because they only investigated CCS, not CCU). The second search query focused on real options studies for $CO_2$-EOR (the second search query was ("EOR" OR "enhanced oil recovery") AND ("real options")). This search query led to 8 and 15 results in the Web of Knowledge and Scopus database, respectively. Of the 23 papers that were retrieved, only nine unique papers were found to fit the scope of this study. Assembling the results from the first and second search query created a literature set of 13 unique studies. Finally, four additional papers that were previously known to the authors for their application of real options in CCU projects were added [17–20]. The selection of the literature set is shown in Figure 1.

As a consequence, a literature set of 17 papers was established. This literature set was screened for several features that were relevant for the application of real options analysis to their study:

- Year and country;
- Type of CCU technology: direct use of $CO_2$ or $CO_2$ conversion;
- Business model: non-cooperative or cooperative;
- Research focus: project valuation (optimal timing or valuing flexibility), policy appraisal and business model comparison;
- Uncertainty source and modelling;
- Type of real options;
- Valuation technique.

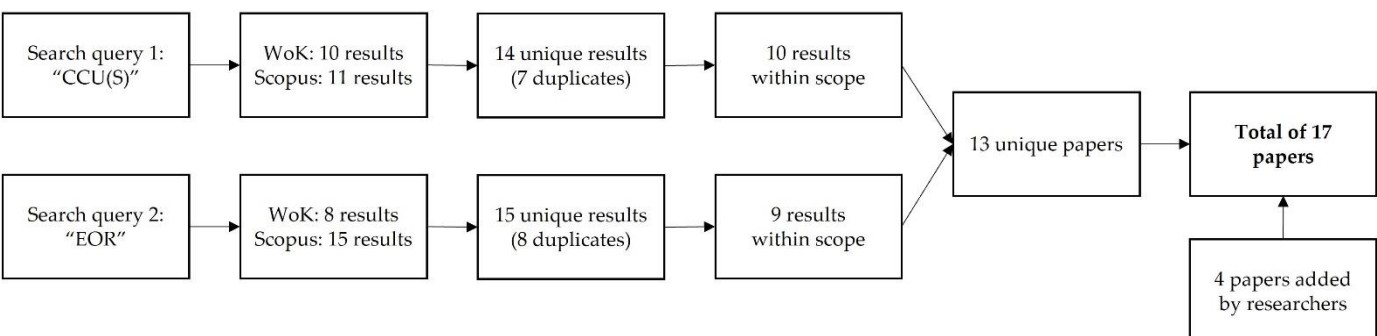

**Figure 1.** Literature set selection.

The features of ROT applications in the CCU literature are discussed in detail in Section 4. Once the existing literature for CCU was screened, the research gaps in these real options studies were evaluated. The following questions were treated in the gap analysis:

(1) What types of uncertainties that are also relevant for CCU projects are not yet addressed?
(2) What types of options were not included in the reviewed literature set although they could be highly valuable for CCU projects?
(3) What valuation techniques are the most suitable to address these research objectives and types of uncertainties?

These potential research gaps were filled in by consulting the basic principles from ROT, as summarized in Section 3. Furthermore, lessons can be learnt from ROT applications in other research areas that have tried to fill these gaps before. Finally, the investment decision sequence is presented for CCU projects, incorporating the real options that can be present and summarizing all sources of uncertainty.

## 3. The Principles of Real Options Theory (ROT)

The term "real options" was claimed for the very first time in 1977 by Myers [21] in a study on the issues of corporate debt. Myers defined real options as "opportunities to purchase real assets on possibly favorable terms" [21] (p. 163).

The opportunities for firms to buy real assets were given this name because of the analogy with financial options. Financial options allow the option holder to buy or sell the financial asset at a fixed price and date in the future. Only if conditions are favorable will option holders exercise their right to buy or sell that financial asset at the agreed-upon terms. Firms with an investment opportunity face a similar dilemma: they have the right, but not the obligation, to invest and acquire the asset in the future [16].

In general, the majority of investment decisions have three characteristics in common: (1) the investment is, at least partially, irreversible, meaning that (part of) the investment cost is sunk; (2) the future returns from the investment are uncertain; and (3) decision-makers have some flexibility in the timing of investment [16]. While the classic net present value (NPV) valuation framework ignores these typical features of investments, the real options theory (ROT) recognizes the ability of decision-makers to adapt the project or technology in response to changes and developments in the real world [10]. In other words, ROT allows for a "now-or-later" decision: investors do not only have to decide whether to invest or not, but also when to invest [13]. Applying ROT to real-life investment decisions (1) acknowledges the fact that the initial costs are (partially) sunk, (2) deals with uncertainties on the future returns of the investment by modelling the future evolution of these uncertainties and (3) introduces various flexibility options into the investment decision, for example, flexibility in the timing of the investment decision, by calculating the value of waiting. In other words, ROT acknowledges the fact that decision-makers have the flexibility to adapt their decisions to changing circumstances, which could improve the potential gains and limit the expected losses of the investment [9]. Moreover, the higher the uncertainty and variability in the payoffs of the investment, the higher the value of

having the option to invest. The intuition is as follows: if a firm has the right—but not the obligation—to invest, more uncertainty and variability in the project payoffs will only increase the potential payoffs from the project whilst leaving the potential losses unchanged (the option to invest will not be exercised at unfavorable conditions) [16].

The remainder of this section expands on the failure of classic NPV valuation techniques, presents the strengths of ROT and lists the main solution methods of ROT.

### 3.1. The Failure of Classic NPV Valuation

Most investment decisions are evaluated by simply calculating the NPV of the project, i.e., the present value of the difference between the revenue and costs. If the NPV is greater than zero, the investment should be made. If the NPV is smaller than zero, the project is expected to lose money and hence, the investment should not be undertaken.

However, this simple NPV rule is based on two implicit assumptions, which may not be valid in a real-life setting. First, the NPV rule implicitly assumes that the investment is reversible, meaning that part of the costs can be recovered if the investment turns out to be less profitable than expected. For reversible investments, the presence of uncertainty does not influence the investment decision: if the project becomes unprofitable later due to unexpected changes, the expenditures can still be recovered [22]. Second, for an irreversible investment, the NPV rule is only valid if the investment decision is a "now-or-never" decision, i.e., you have to decide now whether or not to invest because you will not be able to invest in the future [16]. However, most investment decisions do not meet these conditions. In general, most investments are irreversible to some extent and have the possibility of delaying or staging the investment [16]. To understand why the NPV rule fails in this case, the analogy with financial options is outlined.

Firms facing irreversible investments with the possibility to delay are holding an option: the firm has to right—not the obligation—to invest at some moment in the future. As long as the firm does not make the investment expenditure, it can wait for more information and still change its decision, if desirable. Since waiting allows the firm to collect more information on the future rewards of the investment and reduces the uncertainty of the investment, waiting is valuable to the firm. However, once the investment is made, the firm gives up the possibility to wait and to change its decision if (market) conditions worsen. Hence, investing involves a "lost option value", which should be included as an opportunity cost in the investment decision [16].

Whereas the classic NPV decision rule is to invest when NPV is greater than zero, the improved decision rule should be to invest only when the NPV is greater than the lost option value. ROT provides a framework to value the lost option value or value of waiting and to incorporate it into investment decisions. In sum, ROT adds the time dimension to the decision, thereby making the investment decision dynamic.

### 3.2. Real Options Valuation Techniques

To introduce the time dimension into the investment decision and to include uncertainty and flexibility in the timing of the investment, Dixit and Pindyck [16] outline two mathematical tools: DP and contingent claims analysis. Dynamic programming (DP) splits the whole sequence of investment decisions into two periods: the immediate decision (i.e., now) and the value of all subsequent decisions (i.e., all periods thereafter). By applying recursive optimization methods and comparing the stopping value with the continuation value for each period, the optimal investment decision can be found. Contingent claims analysis is based on the idea of a replicating portfolio: to value a new asset, a portfolio of existing assets is assembled that could replicate the return and risk of the new asset. While DP treats the discount rate exogenously, the contingent claims analysis ensures that the discount rate equals the return the investor could have earned on different assets with similar risks [16].

As ROT became an established approach to investment decisions in various research areas, these valuation techniques were implemented in practice and variations were de-

veloped to value real options. In the current literature, five main solution methods are distinguished for valuing the real options: DP; partial differential equations (PDEs); lattice (or tree-based) models; simulation techniques; and fuzzy set-based approaches [14,15]. These methods can provide analytical or numerical solutions. Analytical solutions are an exact (closed form) solution for the problem, while numerical solutions are found for problems where an exact solution does not exist and the solution has to be approximated [23].

DP is a mathematical recursive optimization method, breaking decisions that span over different periods into sub-problems and finding the optimal decision by working backwards from the last period to the initial decision period [16]. To decide whether to invest now or later, the value of investing now is compared to the continuation value, which is the value of waiting and making the investment in one of the future periods [16]. Intuitively, once the stopping value exceeds the continuation value, it is optimal to invest immediately. This reasoning is also applied in other valuation techniques. At each period, the stopping value and the continuation value are calculated by using one of the below-listed methods, e.g., PDE or Monte Carlo simulations [14]. In sum, DP provides an optimization method to value flexibility options and define the optimal timing of the investment. However, advanced mathematical techniques are needed to solve the problem [24].

The famous Black–Scholes model, which allowed investors to deduce an analytical price for financial options, can also be used to value real options. However, the Black–Scholes equation can only be used to value options with a fixed expiration date (European-type options) and the uncertainty has to follow a stochastic process with constant mean and variance, which is not always realistic. The Black–Scholes equation is just one example of a partial differential equation (PDE) that can be used for real options valuation. The use of PDEs for real options valuation is very accurate and allows for the finding of an analytical solution. However, PDE-based models become analytically unsolvable when more than two uncertainties are involved [12].

Lattice models or trees are probably the easiest and most intuitive models to value real options. Trees are a simple visualization of how the asset can evolve in the future. Lattice models are discrete time models, where the value of the asset is evaluated at each step. The binomial tree model is the most commonly used and most simple lattice model, which has one type of uncertainty and the value of the asset can only take two alternative values at each node (up or down). Lattice models allow the "real options" to be exercised at any chosen time (~American call option), i.e., it evaluates at each node whether the exercise value of the option is greater than the continuation value of the option or not. Lattice models are quicker and more intuitive to grasp, however, they become increasingly complex when more periods and more uncertainties are involved [12]. In sum, lattice models are very effective when only one uncertainty is involved and they can be used to estimate the value of several real options.

Simulation techniques produce distributions with expected values of the project, taking into account the different sources of uncertainty. Monte Carlo simulations allow for different stochastic processes with different probability distributions, thus allowing many different types of uncertainty to be captured in the analysis. Hence, Monte Carlo simulations are suited to solving investment problems with different types of real options and with different sources of uncertainties [15]. However, Monte Carlo simulations only present today's value of the option and are thus incapable of identifying the optimal timing of investment [12]. Monte Carlo simulations are particularly useful for valuing European-type options. However, most real options are American-type options that can be exercised at any time. The features of recursive optimization and DP can be integrated into Monte Carlo simulations to allow the valuation of American-type options as well [25].

The fuzzy sets-based approach is a modern technique to value real options, however, it has not been widely implemented in valuing real options yet [14]. For engineering design or scheduling problems, fuzzy sets are already being used to describe uncertainties and imprecise information for projects [26]. These approaches also model the distribution of value projects. Fuzzy sets preserve some of the advantages of simulation techniques

(e.g., different uncertainties and options), while reducing the computational time requirements [14]. Table 1 summarizes the above-mentioned valuation techniques of real options with their main characteristics.

**Table 1.** The main characteristics of real options valuation techniques, based on [12,14,16,24,25].

| | Dynamic Programming | PDE | Lattice Models | Simulation | Fuzzy Sets |
|---|---|---|---|---|---|
| Numerical/analytical solution | Analytical | Analytical * | Numerical | Numerical | Numerical |
| Outcome | Value of real options + optimal timing | Value of real options | Value of real options | Distribution of project values | Distribution of project values |
| Number of uncertainties | 1–2 | 1–2 | 1 | >1 | >1 |
| Number of options | >1 | 1 | >1 | >1 | >1 |
| Continuous/discrete time model | Continuous | Continuous | Discrete | Continuous | Continuous |
| American-/European-type option | American | European | American | European (and American) | European (and American) |
| (Mathematical) complexity | Low | High | Low | Low | Medium |
| Computation time | High | Low | Low | High | Low |

\* The Black–Scholes model allows the finding of an analytical solution.

### 3.3. Uncertainty Sources

ROT values the managerial flexibility to adjust decisions in response to new information or changing conditions. If everything were 100% certain, new information would not change a project's performance. Hence, the presence of uncertainties that affect a project's performance is a condition for the application of ROT [10]. Even more so, the present uncertainties can be a potential source of value to the project: a higher uncertainty in a project's payoffs increases the value of being flexible and being able to adjust your decisions. In the presence of managerial flexibility, if the uncertainty increases, the potential payoffs from a project also increase while the potential losses will remain the same [16]. Five main sources of uncertainty are identified in the ROT literature:

(1) *Technological uncertainty*: Technological or technical uncertainty describes the uncertainty regarding the amount of time, effort and materials needed to complete a project or regarding the performance of the technology once it is in operation [16]. This uncertainty can only be resolved by actually undertaking the project. However, technical (or endogenous) uncertainties can generally be reduced by active learning [27]. In other words, these uncertainties can be—to some extent—controlled and managed within a project. Examples of technological uncertainties are the level of energy efficiency, raw material consumption and learning effects;

(2) *Market uncertainty*: Market uncertainties refer to the lack of knowledge regarding how a given market will evolve in the future [28]. While technical uncertainties are often endogenous to a project, the majority of market uncertainties are *exogenous*: the source of the uncertainty is external to the project and cannot be controlled or affected by the project. Examples of market uncertainty are product prices, demand uncertainty and electricity prices;

(3) *Policy uncertainty*: Regulations imposed by the government can affect the performance of a project. The prospect of policy changes and the unpredictability of these changes create an additional source of uncertainty [16]. Policy uncertainty refers to the uncertainty that is created when the timing or level of taxes, subsidies or environmental regulations is not yet fixed;

(4) *Impact uncertainty*: Besides providing a product or service, a project also has (unintended) effects on its environment [29]. These effects are called externalities and they are not valued in a market. It is often very difficult to estimate or forecast these effects that projects have on their wider environment, which gives rise to impact uncertainty;

(5) *Societal uncertainty*: Public perception and acceptance can influence the success of a project and create an additional source of uncertainty. For example, wind energy or CCS projects suffered from a lack of social acceptance, which hindered the uptake of these technologies [30].

### 3.4. Real Options

Analogue to financial options, real options refer to the right—but not the obligation—of a firm to undertake a certain investment or acquire a tangible, "real" asset [16]. The higher the degree of uncertainty in a project, the higher the value of having these "real" options, which add the flexibility to respond to uncertain future outcomes. Trigeorgis distinguished six main types of real options [9]:

(1) *Option to delay or defer*: Instead of facing a now-or-never decision, investors can delay their investment to a future period. At each period, they re-evaluate the project and decide whether they should continue (wait for more information), invest immediately or abandon the project completely. The option to delay allows the decision-maker to wait until the uncertainty is resolved or reduced;

(2) *Option to stage (time-to-build investment)*: This option is more subtle than the option to delay. Instead of postponing their investment as a whole, the investment is split up into different phases. First, a partial investment is made, and only when the project's performance meets a certain standard is the second investment undertaken, etc. By staging the investment over time and splitting it into a series of smaller investments, the investor can abandon the project at any stage [9]. This option is particularly important in R&D industries because R&D is typically performed to gain new insights into the technology's performance, which helps the decision-maker to decide whether or not to make further investments [18];

(3) *Option to scale (expand or contract)*: In response to changing market conditions, the scale of production can be either expanded or downsized. To alter the scale of production when a project is already running, additional costs have to be incurred [9]. Firms could also choose to design projects modularly, such that the project could be scaled easily without incurring high additional costs;

(4) *Option to abandon*: This allows managers to (temporarily) shut down the plant when market conditions are weaker than expected. By abandoning the project, additional losses can be avoided [9];

(5) *Option to switch*: This refers to the built-in flexibilities to change the input or output, depending on current market conditions. Firms should be willing to pay a premium for projects with a built-in flexibility to either change input to the cheapest future input or switch output to the most valuable future output [9];

(6) *Option to grow (compound option)*: The growth option refers to early investments that lead the way to future opportunities. While these early projects may not be profitable yet, they may be crucial to unlocking future investments. For example, the infrastructure and experience developed for the early project may serve as a stepping stone for the next generations of that product or process [9]. These growth options are again particularly relevant for R&D-heavy industries.

## 4. ROT in the CCU Literature

As explained in the materials and methods section, a literature search was performed to retrieve all studies in CCU research that implement real options thinking when evaluating investment decisions. This literature search revealed the lack of ROT studies in CCU projects: only 17 studies presented an ROT approach to evaluating the investment decision in a new CCU project. Moreover, the majority of these studies investigated a CCU technology that involves the direct use of $CO_2$. Only two studies were found for a CCU conversion route. Table 2 provides a summary of the ROT studies in the CCU literature. As can be seen from Table 2, this literature set only contained recent studies from between 2014 and 2021. The majority of ROT studies were performed in China. China's high share in coal-fired power plants is responsible for an increased interest in CCU technologies, aiming to mitigate the $CO_2$ emissions from these power plants.

**Table 2.** A summary of ROT applications in the CCU literature.

| References | Direct Use or Conversion | Technology | Year | Location |
|---|---|---|---|---|
| Zhang et al. [31] | Direct use | $CO_2$-EOR | 2014 | China |
| Abadie et al. [32] | Direct use | $CO_2$-EOR | 2014 | Northwestern Europe |
| Compernolle et al. [33] | Direct use | $CO_2$-EOR | 2017 | Northwestern Europe |
| Welkenhuysen et al. [34] | Direct use | $CO_2$-EOR | 2017 | Northwestern Europe |
| Wang and Zang [35] | Direct use | $CO_2$-EOR | 2018 | China |
| Yang et al. [36] | Direct use | $CO_2$-EOR | 2019 | China |
| Fan et al. [7] | Direct use | $CO_2$-EOR | 2019 | China |
| Zhang and Liu [37] | Conversion | Industrial and food utilization | 2019 | China |
| Yao et al. [38] | Direct use | $CO_2$-EOR | 2019 | China |
| Fan et al. [39] | Direct use | $CO_2$-EOR and EWR | 2020 | China |
| Li et al. [19] | Direct use | EWR | 2020 | China |
| Zhu et al. [40] | Direct use | $CO_2$-EOR | 2020 | China |
| Zhang et al. [41] | Direct use | $CO_2$-EOR, ECBM, bio-conversion and chemical synthesis | 2021 | China |
| Compernolle and Thijssen [17] | Direct use | $CO_2$-EOR | 2021 | Northwestern Europe |
| Deeney et al. [18] | Conversion | $CO_2$-to-methane (Sabatier) | 2021 | Northwestern Europe |
| Lin and Tan [42] | Direct use | $CO_2$-EOR | 2021 | China |
| Bi et al. [43] | Direct use | $CO_2$-EOR | 2021 | China |

*4.1. Research Objectives in CCU Projects*

Real options analyses can be implemented for different reasons. In the reviewed literature set, four main research objectives were observed:

(1) *Project valuation—optimal timing (OT) of the investment*: ROT can be applied to value or evaluate the CCU project and to investigate when would be optimal to invest in that CCU technology. In other words, the investment threshold levels of particular economic or technical parameters, such as $CO_2$ price, can be determined. Moreover, the application of real options can help to demonstrate how the presence of different uncertainties affects these threshold levels. Twelve papers from the reviewed literature set investigated the optimal timing of investments by determining the investment threshold levels of $CO_2$ price [17,19,31,33,34,37,39,42], oil price [17,32,33,42] and/or $CO_2$ utilization rate [37]. Yao et al. [38] calculated the investment probability in the CCU project over different stages of the project;

(2) *Project valuation—valuing flexibility (VF)*: Besides determining the optimal timing of the investment, project valuation by real options analysis can also include the valuation of the flexibility that is embedded into the investment decision. The value of the real options themselves was estimated in four studies in the literature set. Zhang et al. [31] valued the cost-saving effect of pre-investing in a carbon capture facility. Abadie et al. [32] explicitly calculated the option value of being able to delay the investment to a later phase. Wang and Zang [35] estimated the value of the compound option by calculating the difference between the static (or passive) NPV and the dynamic NPV (that includes the compound real options). Finally, Deeney et al. [18] benchmarked the real options-based value of the R&D investment against a static NPV that did not include any flexibility;

(3) *Policy appraisal*: The applications of ROT can also be helpful in evaluating how different policy instruments affect the timing or level of investments in CCU projects, subject to uncertainty. To evaluate the effectiveness of different policy incentives, the real options-based value of CCU investments was calculated under different government subsidy modes or levels in seven papers from the literature set [7,33,36,39–42];

(4) *Business model*: The evaluation of different business models can demonstrate how investments in CCU projects can be optimized. The CCU value chain involves different processes, from $CO_2$ capture to $CO_2$ utilization, with different partners being responsible for each process. The business model describes how the avoided carbon taxes, the costs and the revenue from the $CO_2$-based product are distributed among the different stakeholders in the CCU value chain. How these stakeholders cooperate can affect investment decisions in CCU projects. Zhu et al. [40] investigated three

different business models between a coal-fired power plant (CFPP) and a oil producer with different contract terms: a fixed $CO_2$ price; an oil-indexed $CO_2$ price; and a joint venture contract. Compernolle and Thijssen [17] compared the investment thresholds for a CFPP ($CO_2$ capture) and an oil producer ($CO_2$-EOR) on a stand-alone basis with their investment thresholds in a joint venture.

These research objectives are listed in Table 3. The most common research objective was determining the optimal timing of the investment (10 papers), followed by policy appraisal (seven papers). The investigated business models are also listed in Table 3. Non-cooperative investments refer to separate investment decisions by the different stakeholders in the CCU value chain, e.g., the $CO_2$ capture and $CO_2$ utilization plants. When contract terms are settled, some price agreements are made between the different stakeholders. In a joint venture, the CCU partners collaborate and the costs and revenue are distributed between the different stakeholders. In a vertically integrated business model, one investor integrates and operates all steps in the CCU value chain as a whole [20,36,42].

**Table 3.** Research objectives in the CCU literature set.

| References | Research Objective | Business Model | Research Findings |
|---|---|---|---|
| Zhang et al. [31] | Project valuation (OT + VF) | Non-cooperative | Critical $CO_2$ price and investment probability |
| Abadie et al. [32] | Project valuation (OT + VF) | Non-cooperative | Critical oil price |
| Compernolle et al. [33] | Project valuation (OT) + policy appraisal | Non-cooperative + contract terms | Critical $CO_2$ and oil price |
| Welkenhuysen et al. [34] | Project valuation (OT) | Non-cooperative | Critical $CO_2$ price |
| Wang and Zang [35] | Project valuation (VF) | Non-cooperative | Value of compound option and critical $CO_2$ price |
| Yang et al. [36] | Policy appraisal | Vertical integration | Comparison of three subsidy modes and critical oil price |
| Fan et al. [7] | Policy appraisal | Non-cooperative | Comparison of three subsidy modes |
| Zhang and Liu [37] | Project valuation (OT) | Non-cooperative | Critical $CO_2$ price and $CO_2$ utilization rate |
| Yao et al. [38] | Project valuation (OT) | Non-cooperative | Investment probability |
| Fan et al. [39] | Project valuation (OT) + policy appraisal | Non-cooperative | Critical $CO_2$ price |
| Li et al. [19] | Project valuation (OT) | Non-cooperative | Critical $CO_2$ price |
| Zhu et al. [40] | Business model + policy appraisal | Contract terms + joint venture | NPV under different business models and $CO_2$ price |
| Zhang et al. [41] | Policy appraisal | Non-cooperative | Critical government subsidy level |
| Compernolle and Thijssen [17] | Business model + project valuation (OT) | Non-cooperative + joint venture | Critical $CO_2$ and oil price |
| Deeney et al. [18] | Project valuation (VF) | Non-cooperative | Value of compound option and critical $CO_2$ price |
| Lin and Tan [42] | Project valuation (OT) + policy appraisal | Vertical integration | Critical $CO_2$ and oil price |
| Greig and Uden [20] | Project valuation (VF) | Vertical integration | Option value of CCU in transition to net-zero emissions |
| Bi et al. [43] | Project valuation (OT) | Non-cooperative | Unknown (Full paper unavailable to the authors.) |

*4.2. Uncertainty Sources and Modelling in CCU Projects*

Although the CCU landscape is diverse, covering both mature technologies (e.g., $CO_2$-EOR) and emerging technologies (e.g., $CO_2$-based fuels or chemicals) [2], all CCU projects are subject to many different types of uncertainty or risk. The sources of uncertainty that are addressed in the reviewed literature set are listed below, as per the uncertainty

types defined in Section 3.3. Moreover, the techniques used to model the evolution of these uncertainties are summarized as well.

### 4.2.1. The $CO_2$ Price

The major identified source of uncertainty in CCU projects is $CO_2$ price, which remains hard to predict. The evolution of $CO_2$ price in the EU Emissions Trading System (ETS) is driven by both policy pressures and market forces. The $CO_2$ reduction targets set by the EU increase pressure on the carbon market, pushing $CO_2$ price upwards. Other price drivers are oil and electricity prices, which provide signals reflecting the demand for oil and electricity and, consequently, also influence the "demand" for $CO_2$ emissions [44]. As $CO_2$ price is subject to both policy and market forces, it is included as a separate type of uncertainty for CCU projects. The majority of the reviewed studies (13) described the evolution of the price of $CO_2$ using a Geometric Brownian Motion (GBM): the $CO_2$ price follows a non-stationary stochastic process with constant drift and variance [7,17–19,31–33,35,37–39,41,42]. Zhu et al. [40] included $CO_2$ price as an uncertain parameter in a sensitivity analysis and Welkenhuysen et al. [34] treated $CO_2$ price as a stochastic parameter in a Monte Carlo simulation.

The effect of $CO_2$ price uncertainty on the level and timing of CCU investments differs in the literature set. Compernolle et al. [33] found that uncertainty in the price of $CO_2$ emission allowances delays CCU investment. $CO_2$ price and oil price thresholds were higher in the real options-based analysis compared to the traditional NPV approach. This study also showed that lower $CO_2$ price uncertainty reduced the oil price threshold level for $CO_2$-EOR investment. Lin and Tan [42] found that an increase in the volatility of $CO_2$ price (i.e., more uncertainty) leads to a reduction in the value of CCU investments and, hence, delays the investment. Wang and Zang [35], on the other hand, observed that the critical $CO_2$ price is lowered, based on a compound real options model. The threshold may be lowered because investors focus more on the future potential revenue that may follow when considering the compound option. Li et al. [19] concluded that the uncertainty in the $CO_2$ emission rights is the most decisive factor in investment decisions. In this study, a higher uncertainty in the $CO_2$ price induced earlier investment.

### 4.2.2. Technological Uncertainty

Technological uncertainties are particularly important for low-maturity CCU technologies, which are still in the development phase. In the reviewed literature set, four different technological uncertainties were observed: technological progress or learning; residual lifetime; running time; and the EOR recovery factor or EOR efficiency rate.

Eight studies introduced technological learning in the investment analysis [7,18,31,35–37,41,42]. The majority of these studies (seven) modelled technological progress through learning curves, expressing the reduction in investment and/or O&M costs of the project. However, Deeney et al. [18] described the number of technological breakthroughs for an R&D investment using a Poisson process. Li et al. [19] and Zhang et al. [31] investigated the influence of residual lifetime in a sensitivity analysis. Welkenhuysen et al. [34] included the EOR recovery factor as a stochastic parameter in a Monte Carlo simulation. Zhu et al. [40] also investigated the impact of the uncertain EOR efficiency rate. However, Zhu et al. split the evolution of the EOR rate into three periods, and in each period, the EOR rate was described by a different GBM. Remarkably, the uncertainty in the $CO_2$ utilization or $CO_2$ conversion rate was not investigated in the reviewed literature set.

### 4.2.3. Market Uncertainty

Due to the dominance of $CO_2$-EOR as a CCU technology in the literature set, the most common source of market uncertainty was oil price. Six studies characterized the evolution of oil price by a GBM [17,31,33,36,38,40], one study included different levels of oil price in a sensitivity analysis [39], another study included oil price as a stochastic parameter in a Monte Carlo simulation [34] and two studies described the evolution of oil price by a

mean reversion process [32,42]. One study in particular applied the Ornstein–Uhlenbeck model [42], which is the most simple mean reverting process. For commodity prices, this mean reverting process may be more realistic than the GBM, which can wander far from the starting point [16]. Electricity price was treated as an uncertain parameter in three studies, either through a mean reverting process [32] or by including it in a scenario analysis [19,39]. Thirdly, coal price was described by a GBM in [7,35,38]. Fourthly, the price of $CO_2$ that can be used for industrial utilization or food-grade utilization was also described by a GBM in [37]. Finally, the product price was included in one study as market uncertainty. The price of natural gas, serving as a proxy for methane, was described by the Ornstein–Uhlenbeck model in [18].

These market uncertainties can have different effects on the value and timing of CCU investments. Lin and Tan [42] observed that a higher volatility in oil price leads to reductions in CCU investments and delays the investment timing. Compernolle et al. [33] and Abadie et al. [32] found that the oil price threshold is significantly higher when the option to delay is included compared to the traditional NPV approach.

### 4.2.4. Policy Uncertainty

Governments can affect CCU projects by granting subsidies, imposing new $CO_2$ emission reduction targets on the industry or changing the cap on the number of emissions allowances of the EU ETS. As $CO_2$ price is driven by both market and policy uncertainty, it was identified as a separate type of uncertainty. Seven studies from the reviewed literature set investigated the effect of government subsidies on CCU investments by including different modes or levels of government subsidy in a scenario or sensitivity analysis [7,19,31,35,36,39,41].

Zhang et al. [41] calculated the critical government subsidy level to stimulate CCU investment and found that government subsidies alone are not sufficient to incentivize investments in CCU technologies.

### 4.2.5. Impact and Societal Uncertainty

CCU projects may reduce $CO_2$ emissions, having a clear impact on the environment. Arning et al. [30] observed that the perceived risks and benefits of CCU technologies influence the social acceptance rate. However, none of the studies in the reviewed literature set included any type of impact or societal uncertainty.

The uncertainty sources that were included in the CCU literature set are summarized in Table 4.

**Table 4.** Uncertainty sources in real options studies on CCU.

| Uncertainty Source | Number of Ref. | Refs. |
|---|---|---|
| $CO_2$ price | 15 | [7,17–19,31–35,37–42] |
| Oil price | 10 | [17,31–34,36,38–40,42] |
| Learning | 8 | [7,18,31,35–37,41,42] |
| Government subsidy | 7 | [7,19,31,35,36,39,41] |
| Electricity price | 3 | [19,32,39] |
| Coal price | 3 | [7,35,38] |
| EOR rate | 2 | [34,40] |
| Residual lifetime | 2 | [19,31]] |
| Product price | 1 | [18] |
| Running time | 1 | [31] |

### 4.3. Real Options in CCU Projects

In the reviewed literature set, only three different types of options were observed: the option to delay (nine); the option to abandon (two); and the option to grow (two).

In general, investors with an option to delay can choose to invest immediately or postpone the investment to a later period. As many CCU technologies are still in

development, it can be very useful to delay the investment decision and wait until more information is known (on technological performance, market conditions or policy regulations). The option to delay or defer the CCU investment was investigated in 14 studies [7,17,19,31–34,36–39,41–43].

Two studies investigated the option to abandon the CCU project when the economic performance is unfavorable. Zhu et al. [40] assumed that the $CO_2$-EOR operators hold the option to abandon the project if the payoff turns out to be negative. The value of the abandonment option is then equal to the avoided negative cashflows. Welkenhuysen et al. [34] also included the option for $CO_2$-EOR projects to abandon oil fields. The oil production curves of oil fields typically increase at first, reach a peak and then decline again until production is no longer beneficial.

The more complex option to grow or compound was implemented in three studies. Wang and Zhang [35] established a compound real options model to take into account the phased nature of CCS investment decisions from the perspective of a CFPP. Investing in $CO_2$ capture unit opens up the opportunity to invest in a $CO_2$-EOR activity in a second phase. Deeney et al. [18] modelled an R&D investment opportunity as a compound real options structure: at the end of the early phase of the R&D, the decision had to be made to start the late phase of the R&D or not. Yao et al. [38] investigated the investment decision in a coal-to-liquid (CTL) plant, possibly combined with a CCS plant. The decision sequence was split into different phases, where the investor had to decide first whether to build the CTL plant or not, followed by the decision to retrofit the CCS plant or not. If the plants were built, there was still the choice to operate the plants or not. Hence, the investment decisions made in the first stages (building the CTL/CCS plant or not) opened up future growth options.

In the reviewed literature set, no stage, scale or switch options were observed. However, each of these options could be valuable in CCU projects. The option to switch input or output is particularly relevant for CCU projects. Due to the variety in $CO_2$ sources, the ability to switch input would allow CCU projects to use flue gases from different sources with different $CO_2$ concentration levels. The flexibility to switch output from the current product to the most expensive product at that time could also improve the economic feasibility of CCU projects.

### 4.4. Valuation Techniques in CCU Projects

Three main valuation techniques were observed in the reviewed literature set: Monte Carlo simulations (seven); lattices (seven); and dynamic programming (three). The Black–Scholes model or fuzzy sets-based approaches were not applied in the literature set.

The Monte Carlo simulations were used for different research objectives: to determine the optimal timing (five); to value flexibility (two); to compare business models (one); and to evaluate policy instruments (two) (Table 3). The Monte Carlo simulations were also used for the three different types of options: the option to delay (three); the option to abandon (two); and the option to grow (one). Hence, Monte Carlo simulations seem to fit the different research objectives and the different types of options.

The lattice models could also be implemented for different goals: to find the optimal timing (three); to value flexibility (two); and to evaluate policy instruments (four). Both binomial and trinomial trees were built in the literature set. The use of lattice models seems particularly attractive for valuing the option to delay (six) because trees allow an intuitive comparison of different timing alternatives [7,19,39,41]. One study used a binomial tree to value the compound option [35].

Finally, dynamic programming was observed in three studies, for various research objectives: defining the optimal timing (three) and in combination with policy appraisals (one) or business model comparisons (one). In these three studies, dynamic programming was used to investigate the option to delay.

This brief overview demonstrates that different valuation techniques can be used for various research objectives and types of options. Dynamic programming and lattice models

seem to be the most obvious techniques to model the option to delay, while Monte Carlo simulations can be more easily implemented for different types of options.

Table 5 summarizes the types of uncertainties, types of real options and the valuation techniques that were implemented in the reviewed literature set. A capital X refers to a stochastic process, while a lowercase x indicates that the uncertainty was modelled deterministically. The superscript indicates the type of stochastic process: GBM refers to Geometric Brownian Motion; SA to sensitivity or scenario analysis; LC to learning curve models; P to the Poisson process; and OU to the Ornstein–Uhlenbeck model.

Table 5 shows how different uncertainties were combined with different types of options and how different valuation techniques were used to value the same type of option. This demonstrates the wide variety of methods that is currently present in the CCU literature.

**Table 5.** A summary of the uncertainty sources, types of real options and valuation techniques in the reviewed CCU literature set.

| References | Technology | Uncertainty | | | | Real Options | | | Valuation |
| | | $CO_2$ Price | Technological | Market | Policy | Delay | Abandon | Grow | |
|---|---|---|---|---|---|---|---|---|---|
| Zhang et al. [31] | $CO_2$-EOR | X [GBM] | x [LC] | X [GBM] | x [SA] | X | | | Lattice (trinomial tree) |
| Abadie et al. [32] | $CO_2$-EOR | X [GBM] | | X [GBM] | | X | | | Simulation (MC) |
| Compernolle et al. [33] | $CO_2$-EOR | X [GBM] | | X [GBM] | | X | | | DP |
| Welkenhuysen et al. [34] | $CO_2$-EOR | X | X | X | | X | X | | Simulation (MC) and decision tree |
| Wang and Zang [35] | $CO_2$-EOR | X [GBM] | x [LC] | X [GBM] | x [SA] | | | X | Lattice (binomial tree) |
| Yang et al. [36] | $CO_2$-EOR | | x [LC] | X [GBM] | x [SA] | X | | | Lattice (trinomial tree) |
| Fan et al. [7] | $CO_2$-EOR | X [GBM] | x [LC] | X [GBM] | x [SA] | X | | | Lattice (trinomial tree) |
| Zhang and Liu [37] | $CO_2$ conversion | X [GBM] | x [LC] | X [GBM] | | X | | | DP and LSMC |
| Yao et al. [38] | $CO_2$-EOR | X [GBM] | | X [GBM] | | X | | X | Simulation (MC) |
| Fan et al. [39] | $CO_2$-EOR and EWR | X [GBM] | | x [SA] | x [SA] | X | | | Lattice (trinomial tree) |
| Li et al. | EWR | X [GBM] | x [SA] | x [SA] | x [SA] | X | | | Lattice (trinomial tree) |
| Zhu et al. | $CO_2$-EOR | x [SA] | X [GBM] | X [GBM] | | | X | | Simulation (MC) |
| Zhang et al. [41] | $CO_2$-EOR and $CO_2$ conversion routes | X [GBM] | x [LC] | | x [SA] | X | | | Lattice (binomial tree) |
| Compernolle and Thijssen [17] | $CO_2$-EOR | X [GBM] | | X [GBM] | | X | | | DP |
| Deeney et al. [18] | $CO_2$-to-methane (Sabatier) | X [GBM] | X [P] | X [OU] | | | | X | Simulation (MC and random tree) |
| Lin and Tan [42] | $CO_2$-EOR | X [GBM] | x [LC] | X [OU] | | X | | | Simulation (MC) |
| Bi et al. [43] | $CO_2$-EOR | | | | | X | | | |

X: the uncertainty was modelled as a stochastic process; x: the uncertainty was modelled as a deterministic process; [GBM]: the uncertainty was modelled as a stochastic process, described by a GBM; [LC]: a learning curve model described the reduction in investment and/or O&M costs through technological progress; [SA]: the uncertainty was only included as parameter in a sensitivity or scenario analysis; [P]: the technological breakthroughs were described by a Poisson process; [OU]: the oil price/natural gas price (proxy for methane) were described by the Ornstein–Uhlenbeck model.

## 5. Research Gaps and Recommendations

The application of ROT to evaluate investment decisions in novel CCU projects is a relatively new but promising branch in the literature. In this section, the observed research gaps in the previous sections are summarized and several recommendations are made to gain more insights into ROT in future studies. These recommendations are based on real options analyses in other research areas, e.g., CCS, (renewable) energy and climate change policy. First of all, it is remarkable that the vast majority of ROT studies in the CCU literature focused on the direct use of $CO_2$. Only two studies addressed investment decisions for $CO_2$ conversion routes. However, the $CO_2$ conversion routes are also affected by different sources of uncertainty, which restrains firms from investing in these novel CCU projects. Hence, ROT studies can be very valuable in gaining more insights into how these uncertainties affect the optimal investment timing.

### 5.1. Uncertainty Sources

Figure 2 illustrates how the investment decision sequence for CCU projects may look. In Figure 2, all sources of uncertainty are summarized, both for the $CO_2$ capture and the $CO_2$ utilization phases. The uncertainties that were already observed in the reviewed literature set are indicated with asterisks (*).

#### 5.1.1. The $CO_2$ Price

The vast majority of reviewed papers included $CO_2$ price as the main source of uncertainty. The $CO_2$ price uncertainty was included because it affects the revenue from CCU investments. By capturing and utilizing $CO_2$, $CO_2$ emissions into the atmosphere are avoided and thus, the carbon taxes that should otherwise be paid can be avoided as well. Note that this reasoning assumes that the captured and utilized $CO_2$ would be counted as "avoided $CO_2$ emissions" in the carbon trading mechanism. However, in the current EU ETS framework, only permanently stored $CO_2$ counts as "avoided $CO_2$" [45]. Captured $CO_2$ that is used for EOR or other conversion routes falls outside of the current scope of the EU ETS. Hence, it may be insightful to also model the inclusion or exclusion of CCU pathways in the EU ETS framework (or other carbon trading mechanisms) as a source of uncertainty. Even if carbon taxes can be seen as revenue for CCU projects, how carbon price evolution should be described is still up to debate. In sum, two major gaps concerning $CO_2$ price uncertainty have been identified:

- Current feasibility studies do not acknowledge the uncertainty on whether carbon tax would be avoided or not. The uncertainty about the eligibility of CCU in the EU ETS framework (or other carbon trading mechanisms) should be recognized;
- The vast majority of the real options-based studies assumed that carbon prices follow a GBM. However, the behavior of carbon prices in the EU is volatile and prone to jumps [46]. Hence, other stochastic models could be more appropriate;

Several recommendations are made to fill these gaps, based on how carbon price uncertainty is treated in other fields:

- Blyth et al. [47] investigated how uncertain climate change policies affect investment decisions in power generation. In this study, carbon price was used as a proxy for climate change policy. Their model allows the carbon price to change in two distinct ways: (1) carbon prices fluctuate due to changes in demand and supply on the carbon market. This stochastic nature of carbon markets can be described by GBM and (2) the carbon price is also affected by discrete policy-related interventions. These events can be modelled by jump processes, describing the sudden changes in the carbon price levels. The use of a GBM with a jump process to describe $CO_2$ price evolution was also suggested in CCS investment studies [48,49]. Hence, the combination of a GBM to describe the stochastic nature of the carbon price market with a discrete jump process to describe sudden policy changes could be more fitting to describe carbon price evolutions;

- Flora et al. [46] analyzed how price dynamics in the EU ETS affect low-carbon investments. Flora et al. suggest using a more general stochastic model than a GBM because carbon price behavior is still very volatile, uncertain and very prone to jumps. The Variance Gamma has been found to describe the carbon price evolution better than a traditional GBM.

### 5.1.2. Technological Uncertainty

Technological progress or learning was the main identified source of technological uncertainty in the reviewed literature set. Seven (out of eight) studies described the technological improvements using a learning curve model. Besides technological learning, the EOR rate was often included as an uncertain factor as well. Nevertheless, three research gaps could be identified:

- A clear lack of studies investigating technological uncertainties that are specific for $CO_2$ utilization routes was observed. The major challenge for CCU is the high stability of the $CO_2$ molecule, resulting in high energy consumption [6]. Moreover, the technical risks associated with the installation and upscale of CCU plants slow down investments [50]. For the conversion of $CO_2$, in particular, the $CO_2$ conversion rate and the energy efficiency are often decisive parameters. However, none of these parameters were included as a source of uncertainty in the literature set;
- Most CCU technologies (e.g., $CO_2$-based fuels or chemicals) are still novel and emerging [2], requiring further development and pilot-scale projects to demonstrate the feasibility of those projects [51]. Hence, the majority of CCU projects (i.e., CO conversion routes) possess R&D characteristics. As R&D projects are typically characterized by sudden breakthroughs, the use of learning curve models may not be suited for CCU projects that are still in the R&D phase.

Three recommendations are made to address technological uncertainty in CCU projects better in the future:

- Future research should try to map the evolution of the $CO_2$ conversion rate or energy efficiency and find a stochastic model that fits the progress of these parameters better. Alternatively, instead of trying to model these technological uncertainties, real options analysis could also be applied to define the critical thresholds that should be surpassed before investing in a CCU project. In the current literature set, researchers mostly determined the critical $CO_2$ and/or oil price levels (Table 3);
- Instead of using learning curve models, Poisson processes should be used to simulate technological breakthroughs that are typical for R&D projects [18];
- Wang and Yang [52] described how to incorporate flexibility into the management of R&D projects under risk. They observed that technological uncertainty, contrary to market uncertainty, cannot be resolved by waiting. Instead, the uncertainty about the technical performance of the new technology can only be resolved by investing in follow-up stages of the R&D process. This has some implications for the value of the option to delay investment, which we will discuss in Section 5.2.

### 5.1.3. Market Uncertainty

At the $CO_2$ source, which was often an electricity producer or a coal-fired power plant, coal prices and electricity prices were identified as market uncertainties. The $CO_2$ utilization phase often entailed $CO_2$-EOR, hence, oil price was included as market uncertainty in the literature set as well. Two major gaps have been identified concerting the market uncertainty:

- Most $CO_2$ conversion routes have high energy requirements due to the stability of the $CO_2$ molecule [6]. Hence, electricity price can be a dominant factor in the operational expenditures of CCU projects. Electricity price was only included as revenue in the literature set, while it can also be a cost for CCU plants;

- For $CO_2$-EOR CCU projects, oil price was included as market uncertainty in six studies as being the price of the product. In CCU conversion routes, on the other hand, $CO_2$ is generally converted into a chemical or fuel. Although the price of this end product would also influence investment decisions, it was not included as uncertainty in the literature set.

Two recommendations are made on how to treat these market uncertainties in future research:

- Electricity price should be included as a source of uncertainty at the $CO_2$ utilization phase due to the high energy consumption levels of $CO_2$ conversion routes. However, researchers have diverse opinions on how to model the evolution of electricity prices. The review of Agaton [15] regarding CCS projects revealed that both mean reverting processes and GBMs were used to model electricity processes. Kozlova [14] observed a similar debate on electricity prices in real options studies for renewable energy projects. The use of a GBM is borrowed from financial options theory, whereas commodity prices are usually known to have a mean reverting nature [14]. Depending on the level of mean reversion, the use of a GBM may still be appropriate to model electricity prices;
- Although product price could be an important source of uncertainty for CCU projects, this uncertainty was rarely included. For $CO_2$-EOR projects, a GBM was used to describe the evolution of oil price. However, for $CO_2$ conversion routes, many different potential products can be identified whose price does not necessarily follow a GBM. Future research should include product prices as uncertainties for $CO_2$ conversion as well, and should try to find the stochastic model that best describes the product price evolution.

### 5.1.4. Policy Uncertainty

Finally, the only policy uncertainty that was investigated in the literature set was the government subsidy uncertainty, which can refer to subsidies for the $CO_2$ capture plant or the $CO_2$ utilization plant. However, inclusion in or exclusion from the EU ETS framework is an important source of policy uncertainty that should be analyzed as well. Two gaps have been identified concerning policy uncertainty in CCU projects:

- Uncertainty about the level of government subsidy was generally only addressed in a scenario or sensitivity analysis;
- The uncertainty about the eligibility of CCU routes in carbon trading mechanisms was not acknowledged nor addressed in investment analysis for CCU projects.

To address these gaps, two recommendations are made, based on similar challenges in CCS literature:

- Instead of investigating the impact of different subsidy schemes or levels in a sensitivity analysis, the uncertainty about the subsidy level could be modelled through Poisson processes. Subsidy modes depend on policy interventions, which are often discrete in nature. CCS projects are also prone to policy uncertainty and the impact of different subsidy modes, in particular. Hence, Huang et al. [53] modelled the uncertainty of subsidies by a Poisson jump process in their evaluation of a CCS technology to characterize the uncertain timing and level of government subsidies.

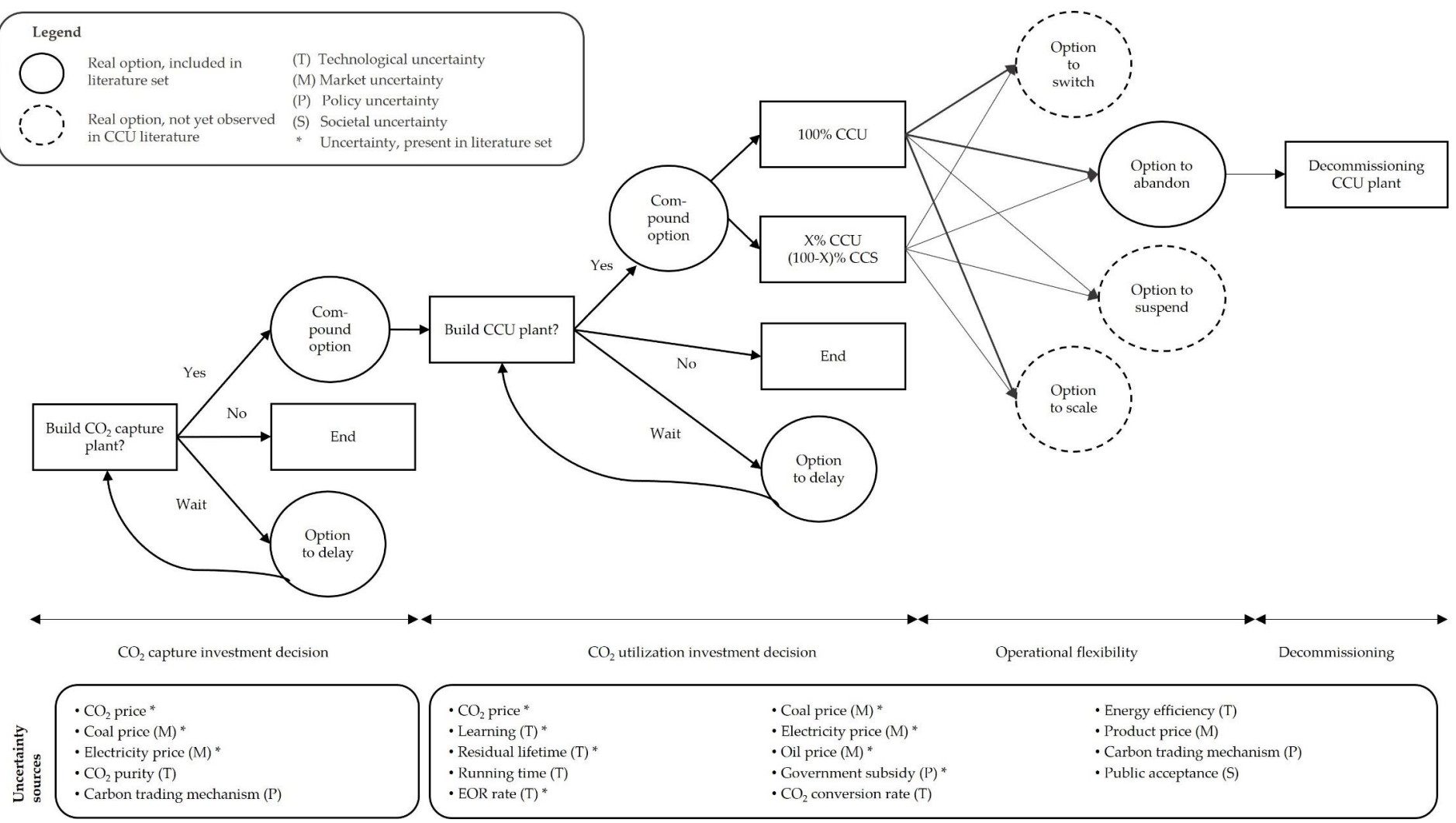

**Figure 2.** Real options in the investment decision sequence for a CCU value chain. Real options that were not yet observed in the literature set are in dashed lines. The sources of uncertainty are listed below. Uncertainty sources observed in the literature set are indicated with a *.

5.1.5. Impact and Societal Uncertainty

Similar to the findings of Agaton [15], who observed the lack of research on social acceptance uncertainty in CCS projects, this review has shown that impact and societal uncertainties were not addressed in the current literature set. Nonetheless, both types of uncertainty could have a substantial impact on the feasibility and desirability of CCU projects. As demonstrated by the public protests against CCS projects, public acceptance is a necessary condition for the successful rollout of CCU technologies [30]. One of the major risks that fuels protests against CCS is the risk of $CO_2$ leakages from the storage reservoir. Narita and Klepper [54] modelled $CO_2$ leakage as a probabilistic event that may occur at a certain hazard rate. Hence, feasibility studies should not only include the economic and environmental impact, but should also address the public acceptance of the CCU technology [30]. Surveys in Germany and the UK have shown that the public is not yet familiar with CCU, but they show a generally positive attitude towards CCU [55,56]. The low public awareness of CCU only increases the uncertainty: will the public acceptance increase further or go down as awareness increases? How this societal uncertainty could be included in investment analysis is an interesting and challenging task for future research.

*5.2. Real Options*

Figure 2 illustrates where the different options are situated in the investment decision sequence for CCU projects. First, the decision has to be made whether to build a $CO_2$ capture now, later or never. Once $CO_2$ can be captured, the investment in a CCU plant can be considered. As the investment in the $CO_2$ capture plant is a prerequisite for the CCU plant to be built, we call this a compound option. The option to delay or defer the investment occurs before the investment is made, while the options to abandon, (temporarily) suspend, switch or scale are only available after the investment has been made. In the reviewed literature set, only three types of real options were analyzed: the option to delay; the option to abandon; and the compound option. The delay or timing option provides the flexibility to invest at the optimal moment. The option to abandon was also present in the literature set. Due to the low maturity of CCU technologies, it can be valuable to be able to abandon the project if the technology does not perform as desired. The compound option, although more complex, was analyzed for three CCU projects. Due to the interlinked steps in the $CO_2$ value chain, investing in one technology may serve as a steppingstone for a novel technology in the next step of the supply chain. The options to scale or switch were not investigated once in the literature set, although both are relevant for CCU projects. Hence, two major gaps have been identified. First, real options-based studies for CCU projects currently lack insights into how the flexibility in the CCU technology itself could add value to a project. For example, the CCU technology could be flexible in terms of input ($CO_2$ source, electricity source) or output (products). This type of flexibility has not yet been investigated or valued in the reviewed literature set. Second, although technical risks associated with the upscale of technologies are one of the main barriers to CCU commercialization, the value of the option to scale up in a later phase has not been analyzed yet. To fill these gaps in the CCU literature, three recommendations are made:

- The flexibility to switch inputs or outputs could be very valuable for CCU technologies. In practice, flue gases from polluting plants are often used as a $CO_2$ source. These flue gases have different compositions and thus, $CO_2$ may be contaminated with nitrogen ($N_2$), methane ($CH_4$), etc. Hence, the input may vary depending on the $CO_2$ source. The option to switch outputs could also be very profitable. If the CCU route could produce a different output with the same input, the products with the highest market value at that moment could be targeted. For example, the option to switch between a higher production of wood chips or the production of energy for sale in a bioenergy cogeneration project was analyzed with a real options approach [57]. The researchers observed that the option to switch added significant value to the project. Moreover, the CCU technology in itself can provide a switch option to the producer: the flexibility to switch between the old polluting production process and the low-carbon CCU

production process. For example, Flora et al. [46] investigated the option to switch electricity production from being fossil fuel intensive to low-carbon sources of energy. In sum, we strongly recommend evaluating the option to switch in future studies as it may increase the desirability of CCU routes;

- The option to scale the production level of CCU plants should be analyzed as well. For example, Enders et al. [58] investigated the option to scale the production level of a natural gas plant. They found that the possibility to scale the plant increases the value of the natural gas significantly;

- Besides these options that can be foreseen in the investment decision sequence of a CCU project, the CCU technology itself can provide flexibility in a system. In the presence of uncertain fossil fuel prices, CCU technologies may offer an advantage because the utilized $CO_2$ often replaces the fossil fuels as a resource in conventional production processes. For example, Davis and Owens [47] applied ROT to estimate the value of renewable energy technologies in the presence of uncertain fossil fuel prices. Renewable energy systems can serve as backup technologies when fossil fuel prices increase severely. Davis and Owens used GBMs to represent the changes in fossil fuel prices and renewable electricity prices and tried to capture the value of the flexibility that renewable energy systems can provide in a real options-based analysis. Similarly, CCU technologies may serve as a backup technology for fossil fuel-based processes when fossil fuel prices rise significantly.

## 6. Conclusions

This review article aimed to explore the application of ROT to introduce and value flexibility in investment decisions regarding CCU projects. CCU projects comprise different phases, from the capture of $CO_2$ to the utilization of $CO_2$, and can unite different agents in the CCU value chain. Hence, many different types of uncertainties and flexibility options can be identified in CCU value chains. Although real options analysis could be very relevant for CCU projects, the literature search revealed the limited number of 17 ROT applications in CCU projects. Moreover, the majority of the literature set focused on the direct use of $CO_2$ (15 studies), while only two studies investigated the investment decisions for $CO_2$ conversion routes. In the reviewed literature set, the price of $CO_2$ was identified as the main source of uncertainty. Oil price (ten), learning effects (eight) and government subsidies (seven) were also frequently observed as uncertainty sources in the literature set. With regard to real options, the option to delay the investment decision was the most common type of option in the literature set. Only two other types of real options were observed in the literature set: the option to abandon and the growth option. Remarkably, the options to switch or to stage were never included, although these could constitute valuable flexibility options for CCU technologies. Several recommendations have been made to improve the insights that can be gained from real options-based analyses for CCU projects. First, the $CO_2$ price uncertainty should not only address the level of $CO_2$ price, but also the eligibility of CCU technologies in carbon trading mechanisms. Second, the technological uncertainty of CCU projects is more than just learning effects. The Poisson process could be useful to model discrete breakthroughs and the uncertainty of the $CO_2$ conversion rate or the energy efficiency of $CO_2$ conversion routes and should also be included. Third, the importance of product price as a market uncertainty should not be neglected for $CO_2$ conversion technologies. Fourth, policy uncertainty should not only be addressed in a sensitivity analyses, but can also be simulated through Poisson jump processes to characterize the uncertain timing and level of government subsidies. Finally, the flexibility embedded into CCU technology should be recognized and valued correctly. For example, the flexibility to switch output or the flexibility to switch from a fossil fuel-based process to a CCU-based process should be included properly in economic feasibility studies. In sum, this review article has shown that a real options analysis can be very valuable for CCU projects to introduce flexibility and uncertainty into investment decisions, thereby making the decisions more realistic. Future research should also apply ROT to

investment decisions for $CO_2$ conversion routes as these are also characterized by different types of uncertainties. Introducing flexibility into the decision pathways of $CO_2$ conversion routes could reduce existing barriers for investors.

**Author Contributions:** Conceptualization, H.L.-V.A.; methodology, H.L.-V.A.; investigation, H.L.-V.A.; writing—original draft preparation, H.L.-V.A.; writing—review and editing, T.C.; visualization, H.L.-V.A.; supervision, T.C. All authors have read and agreed to the published version of the manuscript.

**Funding:** This research was part of the PlasMaCatDESIGN project, which was funded by Research Foundation Flanders (FWO) with grant number S001619N.

**Institutional Review Board Statement:** Not applicable.

**Informed Consent Statement:** Not applicable.

**Data Availability Statement:** Not applicable.

**Conflicts of Interest:** The authors declare no conflict of interest.

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
