# Peer review of "Using Real Options Thinking to Value Investment Flexibility in Carbon Capture and Utilization Projects: A Review"

_sustainability, doi:10.3390/su14042098_

Round 1

Reviewer 1 Report

This study summarizes the application of real options in carbon capture and utilization technology, and the ideas are relatively new and the overall article is relatively clear, but still needs to be explained and modified in the following aspects.

  1. The sample size is too small as there are only 12 relevant papers collected in this study. Is it possible that some of the articles on the application of real options in the field of CCS involve the use of technology and are not adopted in this study. In addition, we found more articles by searching other databases for the keywords provided in the articles, and we suggest that the authors use more databases to expand the sample size and supplement the results.
  2. The real options theory is more widely used in CCS, so it is suggested to clarify why the focus is only on CCU.
  3. The fifth part of the article mentions the application of real options theory in other aspects, and suggests that it is not limited to CCS and R&D.

Reviewer 2 Report

This paper investigates the specific types of uncertainty that arise with the utilization of CO2, identifies the types of real options present in CCU projects, and discusses the applied valuation techniques. This research issue is in line with the current needs of low-carbon sustainable development, and meets the interest of Sustainability in related topics. In general, this manuscript is well done, especially the introduction and the literature analysis. However, I still have several questions to discuss with the authors.

(1) Sections 2 and 5.1

This paper analyzes the investment decision-making problem of real options of CCU (utilization), but why CCS (embedded) is included in the data? It is mentioned that due to some certain similarities of cases of CCS and CCU, it is feasible to learn from the experience of CCS cases. What are the similarities, or what aspects do the similarities  include (such as technical, market, policy, environmental and social)? Moreover, any option investment project may have uncertainties in technical, market, policy, environmental and social aspects. What are the more essential factors? This requires the authors pay attention to the investment-related processes and technical characteristics of CCS and CCU. For example, CCS and CCU may have some similar processes in CO2 capture, transportation and temporary storage. The significance of these similarities for this study needs further explanation. It is suggested that the reasons for adding CCS cases can be better explained in combination with the purpose of this paper.

(2) Section 5.2

It is said in the article that CCU belongs to the R&D project. However, there are many differences among various type of R&D projects. Section 5.2 describes the evaluation methods used in some R&D projects, but does not elaborate on the guiding significance of these R&D projects for CCU projects. Therefore, it is necessary to further explain the similarities and differences among these R&D projects introduced in this paper and CCU projects, and to clarify the reference value and significance of R&D projects more clearly.

Finally, theoretical contributions and innovative insights do not appear do not appear in the conclusion section.

Round 2

Reviewer 1 Report

The article is well revised and the overall structure of the article is relatively clear. The newly added fifth part clearly points out the shortcomings of the current real options model in the application of CCU field and the possible solutions, which is enlightening. It is recommended for publication.